# Multi-Path Recurrent U-Net Segmentation of Retinal Fundus Image

**Yun Jiang †, Falin Wang \*,† , Jing Gao † and Simin Cao †**

College of Computer Science and Engineering, Northwest Normal University, Lanzhou 730070, China; jiangyun@nwnu.edu.cn (Y.J.); 2234254566@gmail.com (J.G.); miss_miette@163.com (S.C.)

\* Correspondence: wangfl0928@gmail.com

† These authors contributed equally to this work.

**Abstract:** Diabetes can induce diseases including diabetic retinopathy, cataracts, glaucoma, etc. The blindness caused by these diseases is irreversible. Early analysis of retinal fundus images, including optic disc and optic cup detection and retinal blood vessel segmentation, can effectively identify these diseases. The existing methods lack sufficient discrimination power for the fundus image and are easily affected by pathological regions. This paper proposes a novel multi-path recurrent U-Net architecture to achieve the segmentation of retinal fundus images. The effectiveness of the proposed network structure was proved by two segmentation tasks: optic disc and optic cup segmentation and retinal vessel segmentation. Our method achieved state-of-the-art results in the segmentation of the Drishti-GS1 dataset. Regarding optic disc segmentation, the accuracy and Dice values reached 0.9967 and 0.9817, respectively; as regards optic cup segmentation, the accuracy and Dice values reached 0.9950 and 0.8921, respectively. Our proposed method was also verified on the retinal blood vessel segmentation dataset DRIVE and achieved a good accuracy rate.

**Keywords:** retinal image segmentation; convolutional neural network; deep learning

## 1. Introduction

According to WHO statistics, approximately 400 million people worldwide have diabetes. Diabetes can cause serious complications such as glaucoma and retinopathy. Effectively segmenting the fundus image assists the physician in diagnosing the disease. The clinical manifestations of diabetic retinopathy are bleeding spots, hard exudation, retinal microvascular abnormalities, new microvascular etc., as shown in Figure 1a,b, which can cause severe visual impairment [1]. As one of the main causes of blindness, glaucoma is irreversible, and it is expected to affect 111.8 million people in 2040 [2]. Glaucoma is usually evaluated in clinical tests by calculating the vertical cup-to-disk ratio (CDR) [3]. The CDR can be obtained by the ratio of the vertical cup diameter (VCD) to vertical disk diameter (VDD). A normal CDR ranges from 0.3 to 0.4; larger CDRs can indicate glaucoma or other ophthalmic neurological diseases (see Figure 1c,d).

The detection of retinal blood vessels is essential for diabetic retinopathy. However, the retinal blood vessel structure is complicated, and the manual segmentation of blood vessels is time-consuming and laborious. At the same time, doctors are required to have extensive professional knowledge. In recent years, with the development of computer technology, many machine learning methods have appeared to detect retinal blood vessels. Azzopardi et al. used a B-Cosfire filter to perform the automatic segmentation of the vessel tree in response to vessels [4]. the authors an automated retinal blood test method combining the topological segmentation of blood vessels and morphological blood vessel extractors was proposed in [5]. Bankhead et al. [6] developed a wavelet transform method to enhance the foreground and background for fast blood vessel detection. the authors constructed a 41-D

feature vector for each pixel and trained an AdaBoost classifier to classify each pixel in the retinal image in [7]. Dharmawan et al. [8] used a direction-sensitive improved matched filter bank to segment the fundus vessels. Deep convolutional neural networks (DCNN) show strong robustness and efficiency in segmenting blood vessels. A new formulation of DCNN that allowed the simple and accurate segmentation of the retinal vessels. A major modification was the reduction of the intra-class variance by formulating the problem as a three-class problem that differentiates large vessels, small vessels, and background areas [9]. Dasgupta et al. described the segmentation task as a multi-label inference task, and used a combination of a convolutional neural network and structured prediction to achieve retinal blood vessel segmentation in [10]. A recurrent convolutional neural network (RCNN) based on U-Net and a recurrent residual convolutional neural network (RRCNN) based on U-Net, which were called RU-Net and R2U-Net, respectively, to solve the problem of blood vessel segmentation was proposed in [11]. A method of connection-sensitive consideration U-Net (CSAU) was proposed to accurately segment retinal vessel in [12].

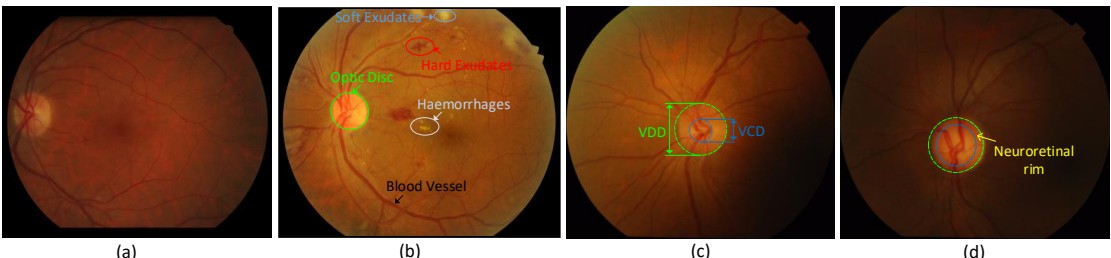

**Figure 1.** (**a**) is a healthy fundus image; (**b**)is an image of diabetic retinopathy with a large amount of bleeding and exudates on the retina; (**c**) is a normal fundus image, and the ratio of the vertical cup diameter (VCD) to vertical disk diameter (VDD) is relatively small; (**d**) is a glaucoma image with a narrow nerve-retinal margin band between the optic disc and optic cup.

The detection of glaucoma can be obtained by CDR, so it is of great significance to segment the optic disc (OD) and optic cup (OC). The existing methods for segmenting the OD and OC mainly include shape-based template matching, active contour and deformable models, and deep learning methods. A new method for automatic disc positioning and segmentation was proposed in [13], The localization process combined vascular and luminance information to provide the best estimate of the center of the optic disc. A new template-based method was proposed to segment the optic disc from digital retinal images [14]. A region growing technique based on an adaptive threshold was proposed for video disc segmentation [15]. Saradhi et al. [16] proposed a novel active region-based active contour model for OD segmentation. The model combined image information at points of interest in multiple image channels to resist changes in the OD region and its surrounding regions. A deformable model based on the active video disc was explored in [17], An active optic disc contains a pair of concentric inner and outer circles, respectively, corresponding to the OD boundary and the local background, which are used to define the local contrast energy. A deformable model was used which incorporated statistical shapes. A set of landmark points on the OD is initialized with the average shape in the training image and iteratively adapts to the test image, which is consistent with the point distribution model representing the shape encountered during the training process [18,19]. Singh et al. proposed a retinal image segmentation method based on a conditional generative adversarial network (cGAN) in [20]. A novel patch-based output space adversarial learning framework (pOSAL) which could jointly and robustly segment OD and OC from different fundus image datasets was proposed in [21] . Sevastopolsky et al. [22] proposed a universal video disc and cup automatic segmentation method based on deep learning, which was a modification of the U-Net convolutional neural network.

The fundus retinal image has a complex structure and low contrast; in particular, the lesion image contains a large amount of bleeding, micro tumors, and hard and soft exudates, which has a great impact on the segmentation of retinal blood vessels and the definition of the optic disc cup boundary. The methods proposed in the above literature usually have the following problems: (1) due to the large

downsampling factor, the single-path downsampling process results in the loss of a large amount of feature information in the retinal image, which ultimately cannot be recovered; (2) insufficiently close cross-layer feature information contact leads to an insufficient understanding of the information—due to the single feature information of the same layer, the feature information cannot be complementarily fused, and ultimately a large amount of feature information is lost; (3) the feature extraction ability of the network structure is insufficient and it is difficult to restore low-level detailed feature information, which generates a great deal of noise in the segmented image; (4) the retinal blood vessel segmentation accuracy is low and the optic disc optic cup boundary segmentation is blurred. In view of the above problems, this paper proposes a multi-path recurrent U-Net architecture to achieve the segmentation of the optic disc optic cup. At the same time, the effectiveness of this method is verified on the retinal vessel segmentation data set DRIVE, and good results are achieved. The main contributions of this work are summarized as follows:

(1)  A multi-path recurrent U-Net network architecture combining a convolutional neural network and a recurrent neural network is proposed . As shown in Figure 3a, this method implements the segmentation of retinal medical images.

(2)  Multi-branching the encoding path and the decoding path can more effectively obtain different semantic features.

(3)  The multi-path recurrent U-Net combines the recurrent neural network to sequence the multi-path output features in time to further improve the target features.

(4)  For the Drishti-GS1 dataset, the method for the segmentation of the OD and OC achieved state-of-the-art results. Our method has also been verified on the retinal blood vessel segmentation dataset DRIVE and achieved a good accuracy rate.

The organizational structure of this article is as follows: Section 2 mainly discusses related work; in Section 3, the multi-path recurrent U-Net model proposed in this paper is described; Section 4 mainly describes the data set, experiments and results; and finally, Section 5 summarizes this article.

## 2. Related Work

### 2.1. U-Net Model

The full convolutional neural network [23] proposed by Jonathan Long can accept input images of any size and uses a deconvolution layer to up-sample the feature map of the last convolution layer to restore it to the same size as the input image. Therefore, a prediction can be generated for each pixel while retaining the spatial information in the original input image, finally performing pixel-by-pixel classification on the up-sampled feature map. Based on fully convolutional networks (FCNs), Ronneberger et al. proposed the U-Net model [24], which has achieved great success in the field of cell segmentation in microscopic tissue sections, as the U-Net achieves acceptable results.

The model has been widely used in various applications of semantic segmentation, such as for satellite image segmentation and industrial defect detection. The use of data augmentation can train some relatively small samples of data, especially medical-related data. The emergence of U-Net has been very helpful for deep learning for medical images with fewer samples. U-Net consists of an encoding path and a decoding path. The encoding path gradually reduces position information and extracts abstract features through the pooling layer. For accurate positioning, the local pixel features extracted from the decoding path will be combined with the new feature map during the upsampling process to retain some important feature information from the previous downsampling process to the greatest extent (see Figure 2a). In 2016, Çiçek et al. proposed that the 3D U-Net model [25] is a simple extension of U-Net and can be applied to 3D image segmentation. The U-Net proposed in [26] is essentially a deeply supervised encoder/decoder network, in which the encoder and decoder subnetworks are connected through a series of nested, dense skip paths. The redesigned skip path aims to reduce the semantic gap between the feature maps of the encoder and decoder subnets.

Oktay et al. [27] introduced an attention mechanism into U-Net. Before stitching the features at each resolution of the encoder and the corresponding features in the decoder, an attention module was used to readjust the output features of the encoder. Because U-Net's encoder/decoder structure and skip connection are a very classic design method, many papers have continued the core idea of U-Net, adding new modules or incorporating other design concepts, and all have achieved excellent results.

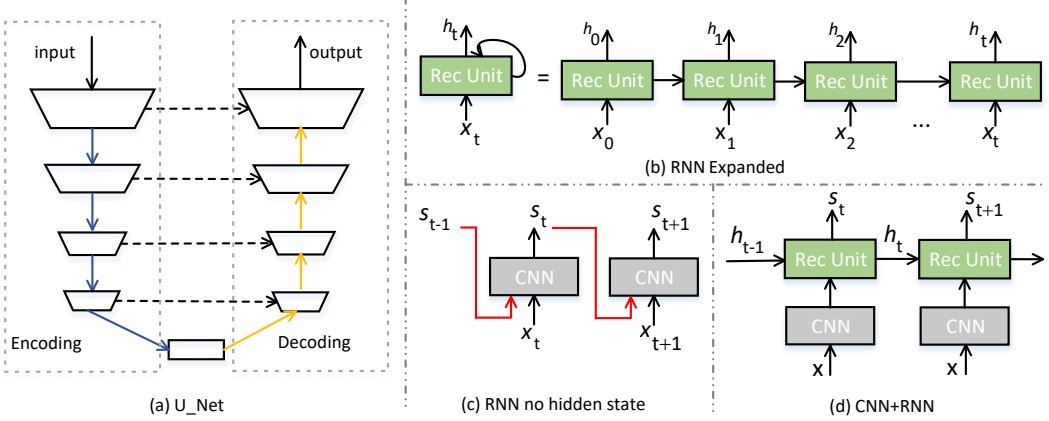

**Figure 2.** The structure of the U-Net, the recurrent neural network (RNN), and the convolutional neural network (CNN) and RNN in combination. (**a**) is the U-net structure; (**b**) is the expended structure of the recurrent neural network; (**c**) is the recurrent neural network without hidden state; (**d**) is the structure of the combination of convolutional neural network and recurrent neural network.

## *2.2. Recurrent Neural Network*

A recurrent neural network is a type of recursive neural network that take sequence data as an input, performs recursion in the direction of the sequence, and connects all nodes in a chain [28] (see Figure 2b). Recurrent neural networks have certain advantages when learning the nonlinear characteristics of sequences. Therefore, they are widely used in the field of natural language processing. Graves et al. used deep recurrent neural networks for speech recognition in [29], and the DiSAN network proposed in [30] combined an attention mechanism with a recurrent neural network for natural language processing. At the same time, the combination of the recurrent neural network and convolutional neural network can effectively deal with computer vision problems including sequence input. Mosinska et al. [31] passed the modified U-Net generated segmentation results back to the network along with the original image as input, which led to the progressive refinement of the segmentation results, as shown in Figure 2c. A novel end-to-end network structure which was based on recursive neural networks to find objects and their segments one by one in turn and learn how to segment instances in order was proposed in [32], as shown in Figure 2d. Wang et al. [33] proposed an effective recursive semantic segmentation method, which could be run in an environment in which the amount of training data and computing power is limited. There is an increasing amount of research into combining the convolutional neural network and recurrent neural network in semantic segmentation. In this paper, U-Net is multi-path processed, combined with recurrent neural unit, and the output features of each path are serialized and passed through the recurrent unit.

## 3. Methodology

In this section, we will introduce the overall architecture of the multi-path recurrent U-Net model proposed in this article in detail. At the same time, the details of the recurrent unit proposed in this article will also be explained.

### 3.1. Multi-Path Recurrent U-Net

The proposed architecture is based on the U-Net model architecture, as shown in Figure 3a. The proposed network structure has a coding path and a decoding path, with skip connections between the corresponding coding layer and decoding layer, thereby allowing the network to retain low-level features for final prediction. In the decoding path, the proposed method resembles a full binary tree. Convolutional blocks are placed at the nodes of each layer. As shown in Figure 3b, the connection line between nodes represents the downsampling pooling operation. After each pooling layer in the encoder, the channel number is doubled. After the input image passes through the first layer of convolution blocks, the maximum pooling operation and the average pooling operation are performed, respectively, corresponding to the left and right branches in the binary tree. The coding path has four layers of convolutional blocks. According to the distribution of full binary tree nodes, the numbers of convolutional blocks in each layer are $L_1 = 1$, $L_2 = 2$, $L_3 = 4$ and $L_4 = 8$. The authors [34–36] showed that the error of feature extraction mainly results from two aspects: (1) the variance of the estimated value increases due to the limitation of the neighborhood size; (2) the error of the convolution layer parameter causes the deviation of the estimated mean. The average pooling can reduce the error of the increase in the variance of the estimated value caused by the limited size of the neighborhood and retain more background information of the image. The maximum pooling can reduce the deviation error of the estimated mean caused by the error of the layer parameters and retain more texture information. The proposed method introduces both types of pooling operations, using the maximum pooling operation on the left branch and the average pooling operation on the right branch to reduce the error. The decoding path relies on transposed convolution to increase the representation ability of the model, and the structure is similar to the encoding path. In each convolution block, since the proposed method usually relies on very small batch sizes, we use group normalization in all convolutional layers. The proposed method serializes the output feature maps of the eight convolution blocks in the fourth layer of the coding path. The proposed method serializes the output feature maps of the eight convolution blocks in the fourth layer of the coding path. First, the output feature of the convolution block at the leftmost leaf node is taken as the start data of the sequence * 1, and then the output feature of the convolution block at the rightmost leaf node is taken as the end data of the sequence * 8. By recursing the sequence, $s_t$ can simply concatenate the previous segmentation mask $s_{t-1}$ with the input image $x_t$ on each recurrent iteration $t$. The resulting cascade tensor is then implemented by a recurrent unit, and finally, the output features of each recurrent unit are input to a decoder in a corresponding decoding path. Next, we will discuss two variants of the model's recurrent unit.

### 3.2. Recurrent Unit

The recurrent unit design in this method is inspired by the gate recurrent unit (GRU) [37]. The encoder represents the input image as a fixed-length vector. Since the proposed method is a multi-path encoder, the vectors output by each path can be serialized. The sequence is then filtered through a recurrent unit to output a new sequence vector, and the encoder uses these sequence vectors to reconstruct the target image. Therefore, this article modifies the GRU equation accordingly, but retains the original purpose of the GRU, as shown in Figure 3c.

In the *t-th* iteration, this article first obtains the two gating states by the previous hidden state $h_{t-1}$ and the current node input $x_t$, resets the gate and updates the gate; see Equations (1) and (2).

$$r_t = \sigma(conv([x_t, h_{t-1}])) \tag{1}$$

$$z_t = \sigma(conv([x_t, h_{t-1}])) \tag{2}$$

where $r_t$ indicates reset gating and $z_t$ indicates update gating. $\sigma()$ represents the sigmoid activation function. Through this function, the data can be transformed into a value in the range of 0–1, thereby serving as a gating signal. [] represents the concatenation of two vectors. $conv()$ represents a $1 \times 1$

convolution operation. The initial weights of $conv()$ in these two gates are different. After obtaining the reset gating information, we aim to generate a candidate update $f_t$ through the tensor combination of $r_t$ and hidden state $h_{t-1}$ with the current input $x_t$. At this time, $f_t$ contains the current input $x_t$, and $f_t$ is added to the current hidden state in a targeted manner, and the information at the current time is retained; see Equation (3).

$$f_t = tanh(conv([x_t, (r_t \odot h_{t-1})]))$$

(3)

where $tanh()$ represents the Tanh activation function, which shrinks the data to the range of $-1 \sim 1$. $\odot$ indicates that the corresponding elements in the operation matrix are multiplied, so the two multiplication matrices are required to be of the same type. The hidden state $h_t$ at the current moment is obtained according to the information of the update gate and the candidate update $f_t$. The operation of this step is performed to forget some of the dimensional information passed by $h_{t-1}$ and add some dimensional information input by the current node. This is shown in Equation (4).

$$h_t = f_t \odot z_t \oplus h_{t-1} \odot (1 - z_t)$$

(4)

where $\oplus$ represents the matrix addition operation.

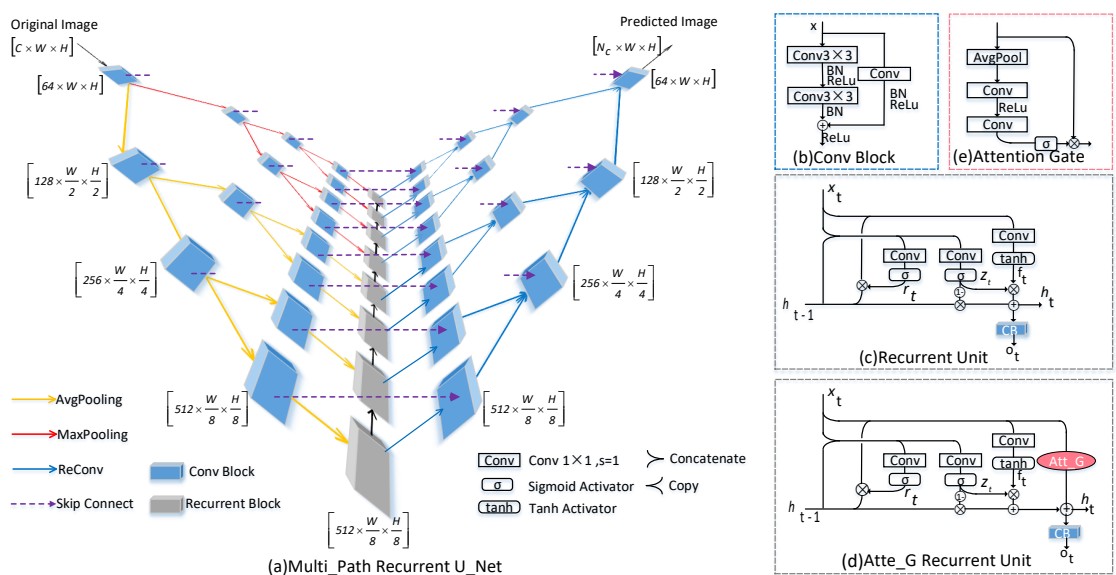

**Figure 3.** Multi-path recurrent U-Net structure and detailed internal structure of the model. (**a**) is the Multi-path recurrent U-Net structure; (**b**) is the basic convolution block; (**c**) is recurrent unit; (**d**) is a recurrent unit with an attention gate; (**e**) is the structure of the attention gate in the recurrent unit.

Finally, according to the hidden state $h_t$ at the current moment, we input $h_t$ to the basic convolution block to obtain the output feature map of the $t - th$ recurrent unit as the input of the corresponding position decoder, as shown in Equation (5).

$$o_t = CB(h_t)$$

(5)

$CB()$ represents the basic convolution block used in this paper. In contrast to GRU, the RU in the proposed method uses $1 \times 1$ convolution in each gate to further abstract the feature, and uses the current concealed state as the output of the recurrent unit through the basic convolution block. Compared with the original U-Net, this allows us to extract a large number of features without having to increase the number of channels in the encoding and decoding layers.

### 3.3. Attention Recurrent Unit

The recurrent unit generates the hidden state at the current moment by gating the current input and the hidden state of the previous recurrent unit. As can be seen from the previous section, the recurrent unit is sensitive to local information but lacks access to multiple global information. Therefore, the proposed method makes further improvements to the recurrent unit. Referring to Figure 3d, the attention gating unit is introduced to obtain global information of the current input. The attention gating unit is shown in Figure 3e.

Specifically, the attention circulation unit and the circulation unit have similar structures, and only the attention gating unit is introduced later. Therefore, the equation is basically the same as above, except for the hidden state. it is now expressed as Equation (6). The attention gating unit first globalizes the current input $x_t$ and turns each two-dimensional feature channel into a real number $r \in \mathbb{R}$. $r$ has a global receptive field to some extent, and the number of channel outputs is the same as the number of channels of the original feature map. Secondly, $R$ is rescaled to $R'$ through two layers of $1 \times 1$ convolution, and then $R'$ is output to $R''$ through the sigmoid activation function. Finally, we multiply $x_t$ and $R''$ as the output at of the gating unit.

$$h_t = (f_t \odot z_t \oplus h_{t-1} \odot (1 - z_t)) \oplus a_t \tag{6}$$

This simple modification combines global information with local information to minimize the tension between semantics and location. As a result, more accurate predictions can be made, and the experiments in this article have also proven this.

## 4. Experiment

### 4.1. Data Set and Data Preprocessing

The Drishti-GS1 dataset [38] contains a total of 101 fundus images: 31 normal and 70 diseased. The training set contains 50 images and the test set contains 51 images. All image OD and OC regions were marked by four ophthalmologists with different clinical experience. In this paper, the average area marked by four experts is used as the standard optic disc and cup area for training, and the pictures have different sizes, as shown in Figure 4a,b. This paper draws on the pre-processing method in [39] to pre-process the original image through polar transformation, which can effectively improve the segmentation efficiency. Due to the small sample size of the Drishti-GS1 data set, in order to prevent the model from overfitting and improve the accuracy and robustness of the model, this paper augments the data of the training set, including the use of a method based on a video disc center point image detection [40] method to extract 10 different sizes of pictures by performing a random horizontal flip, random vertical rotation, and random angle rotation. However, when training the network, the size of the input image is expanded or scaled to the standard $512 \times 512$, the batch size is 4, and the iteration period is 400. Only $700 \times 700$ pictures are extracted on the test set, then scaled to $512 \times 512$ and input into the network, and then the generated predicted image is filled to the original image size.

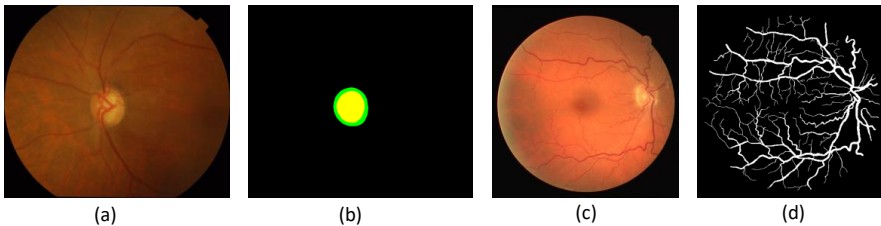

|     |     |     |     |
| --- | --- | --- | --- |
| (a) | (b) | (c) | (d) |

**Figure 4.** Original images and real labels in Drishti-GS1 and the DRIVE dataset. (**a**) is the image number 11 in the Drishti-GS1 dataset; ((**b**) is the ground truth corresponding to (**a**); (**c**) is the image number 22 in the DRIVE dataset; (**d**) is the ground truth corresponding to (**c**).

The DRIVE dataset consists of 40 retinal fundus blood vessel images, corresponding real labeled images, and corresponding masks images. The images originate from a diabetic retinopathy screening program from the Netherlands, and each image has a size of $565 \times 584$. As shown in Figure 4c,d, for the DRIVE data set, during the training process, $10,480$ image patches of size $128 \times 128$ are randomly extracted for the training set. At the same time, $10,480$ real label patches with a size of $128 \times 128$ are extracted from the corresponding real labels at the same location to calculate the loss and train the network. The input image batch size is 16 and the iteration period is 200. In the test phase, this article extracts image patches for each test picture of each dataset in a sliding window order. The size of the sliding window is $128 \times 128$, the sliding step is 5 pixels, and the part of the sliding window beyond the picture is filled with 0. Similarly, the batch size of the image input to the network is 16.

### 4.2. Implementation Details

The method was built with the open source deep learning library PyTorch, implemented on a server configured with an Inter (R) Xeon (R) E5-2620 V3 2.40GHz CPU, Tesla K80 GPU, and Ubuntu64 operating system. During the training phase, the Adam optimizer was used. The parameters were set to $\beta_1 = 0.9$, $\beta_2 = 0.999$, and $\varepsilon = 1e - 8$. The learning rate $l_r$ was initialized to 0.001, and the learning rate was attenuated by the Plateau [41] method. The loss function used a cross-entropy loss function. The definition is as follows:

$$Loss_{ce}(y, \hat{y}) = -\sum y_i log \hat{y}_i + (1 - y_i) log(1 - \hat{y}) \tag{7}$$

where $y_i$ indicates a true label and $\hat{y}_i$ indicates a predicted image.

### 4.3. Evaluation Indicators

In order to evaluate the effectiveness of the method proposed in this paper on the segmentation of these two data sets, this paper analyzes the sensitivity, specificity, accuracy and F-measure evaluation indicators by generating a confusion matrix.

It should be noted that the F-measure indexed here is the same as the dice score in the disc and cup segmentation fields. In addition, in the evaluation of the Drishti-GS1 data set, this paper adds an evaluation index which is widely used in the industry: boundary distance localization error (BLE) [42]. BLE is used to measure the error (in pixels) between the predicted cup boundary ($C_o$ ) and the true label boundary ($C_g$), which is defined as follows:

$$BLE(C_g, C_o) = \frac{1}{n} \sum_{\theta=0}^{\theta_n} |r_\theta^g - r_\theta^o| \tag{8}$$

where $r_\theta^g$ represents the radial Euclidean distance from the predicted boundary point to the centroid in the direction $\theta$, and $r_\theta^o$ represents the radial Euclidean distance from the boundary point of the real label to the centroid in the direction $\theta$. In the evaluation, 24 equidistant points are considered $n = 24$. The ideal value for BLE is 0.

### 4.4. Comparison of Model Improvement Results

In order to verify the effectiveness of the multi-path U-Net recurrent block (RU) and attention recurrent block (ARU) proposed in this paper, experiments were performed on the Drishti-GS1 dataset and DRIVE dataset. The indicators in the following tables are averages and standard deviations. MPU stands for multi-path U-Net, MPR stands for multi-path recurrent U-Net, and MPAR stands for multi-path recurrent U-Net with an attention gate added to the recurrent block.

In this paper, Drishti-GS1 compares the results of the segmentation of the optic discs of each model, as shown in Tables 1–3. It can be seen that the performance of the MPR model with the recurrent block is improved compared to the basic MPU; in particular, for the $Cup_{Dice}$ indicator, the MPR model

is 3.44% higher than the MPU model, which shows that combining the convolutional neural network and the recurrent neural network is significant for the OC segmentation.

**Table 1.** Comparison of optic cup (OC) model changes based on the Drishti-GS1 dataset. MPU: multi-path U-Net; MPR: multi-path recurrent U-Net; MPAR: multi-path recurrent U-Net with attention gate.

| Model | Accuracy | Specificity | Sensitivity | Dice |
|-------|----------|-------------|-------------|------|
| MPU | 0.9935/0.0037 | 0.9971/0.0024 | 0.7849/0.1558 | 0.8499/0.1194 |
| MPR | 0.9945/0.0030 | 0.9972/0.0019 | 0.8363/0.1507 | 0.8843/0.0837 |
| MPAR | **0.9950/0.0022** | **0.9981/0.0022** | **0.9365/0.0659** | **0.8921/0.0793** |

**Table 2.** Comparison of optic disc (OD) model changes based on the Drishti-GS1 dataset.

| Model | Accuracy | Specificity | Sensitivity | Dice |
|-------|----------|-------------|-------------|------|
| MPU | 0.9953/0.0032 | 0.9971/0.0035 | 0.9155/0.0861 | 0.9724/0.0157 |
| MPR | 0.9964/0.0022 | 0.9978/0.0024 | **0.9390/0.0590** | 0.9737/0.0137 |
| MPAR | **0.9967/0.0022** | **0.9980/0.0020** | 0.9365/0.0639 | **0.9817/0.0156** |

**Table 3.** Comparison of optic model changes based on the Drishti-GS1 dataset.

| Model | Accuracy | Specificity | Sensitivity | Dice |
|-------|----------|-------------|-------------|------|
| MPU | 0.9980/0.0012 | 0.9985/0.0009 | **0.9838/0.0220** | 0.9690/0.0168 |
| MPR | 0.9983/0.0011 | 0.9989/0.0008 | 0.9810/0.0222 | 0.9726/0.0138 |
| MPAR | **0.9985/0.0011** | **0.9990/0.0009** | 0.9811/0.0215 | **0.9740/0.0145** |

To further prove the improved performance of the model, we propose a hypothesis: the MPAR model performs better than the MPU model and MPA model. We conducted P value analysis on Dice, and the results are shown in Table 4. MPAR:MPU represents the comparison between the MPAR model and the MPU model, and MPAR:MPR represents the comparison between the MPAR model and the MPU model. In the table, MPAR:MPU has a $P_{value} < 0.05$ for OC and OD, indicating that the MPAR model and the MPU model have statistical significance; that is, the MPAR model performs better than the MPU model. In the table, MPAR:MPR has a $P_{value} < 0.05$ for OD, indicating that the MPAR model performs better in OD segmentation than the MPR model.

**Table 4.** Comparison of $P_{value}$ analysis results based on Dice.

| | Cup | | Disc | |
|---|-----|-----|------|-----|
| | **MPAR:MPU** | **MPAR:MPR** | **MPAR:MPU** | **MPAR:MPR** |
| $P_{value}$ | 0.038 | 0.630 | 0.003 | 0.007 |

Figure 5 shows the result of segmentation of the OC by different models, and the OC region is enlarged and displayed. The first line of the original image corresponds to the true label of the OD and the corresponding enlarged image, and the remaining lines are the segmentation results images of different models. It can be seen from the enlarged image that the contour of the MPAR model is more consistent when the OD is segmented, but the segmentation contour of the OC is insufficient, and the MPAR model is better at segmenting the OC.

Table 5 shows the BLE errors of the segmentation results of each model. As can be seen from the table, the results of the MPAR model are better than the MPU model and the MPR model. Especially for the video cup, comparing MPAR with MPU, the BLE error is reduced by 6.242 pixels, which also shows that the model in this paper can segment the video cup more accurately. BLE error plays a crucial role in the evaluation of the model. This paper shows that the BLE was 360° (each 15°). Figure 6 shows the average error, median, maximum and minimum error, and quartile box of each test picture in the same orientation.

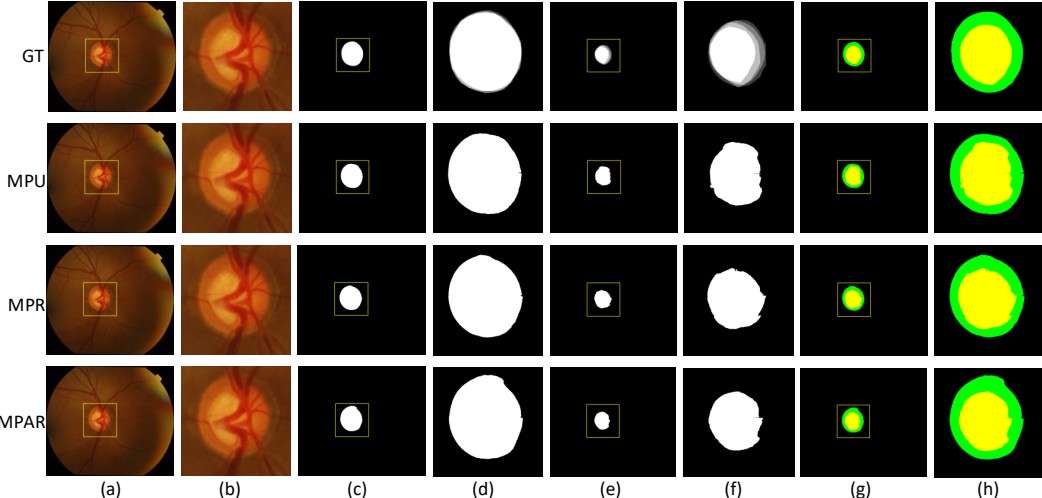

**Figure 5.** Comparison of segmented images of each model in the Drishti-GS1 dataset. (**a**) original image; (**b**) is a partial cut of the original image; (**c**) Disc; (**d**) is a partial cut of the Disc; (**e**) Cup; (**f**) is a partial cut of the Cup; (**g**) Optic; (**h**) is a partial cut of the Optic.

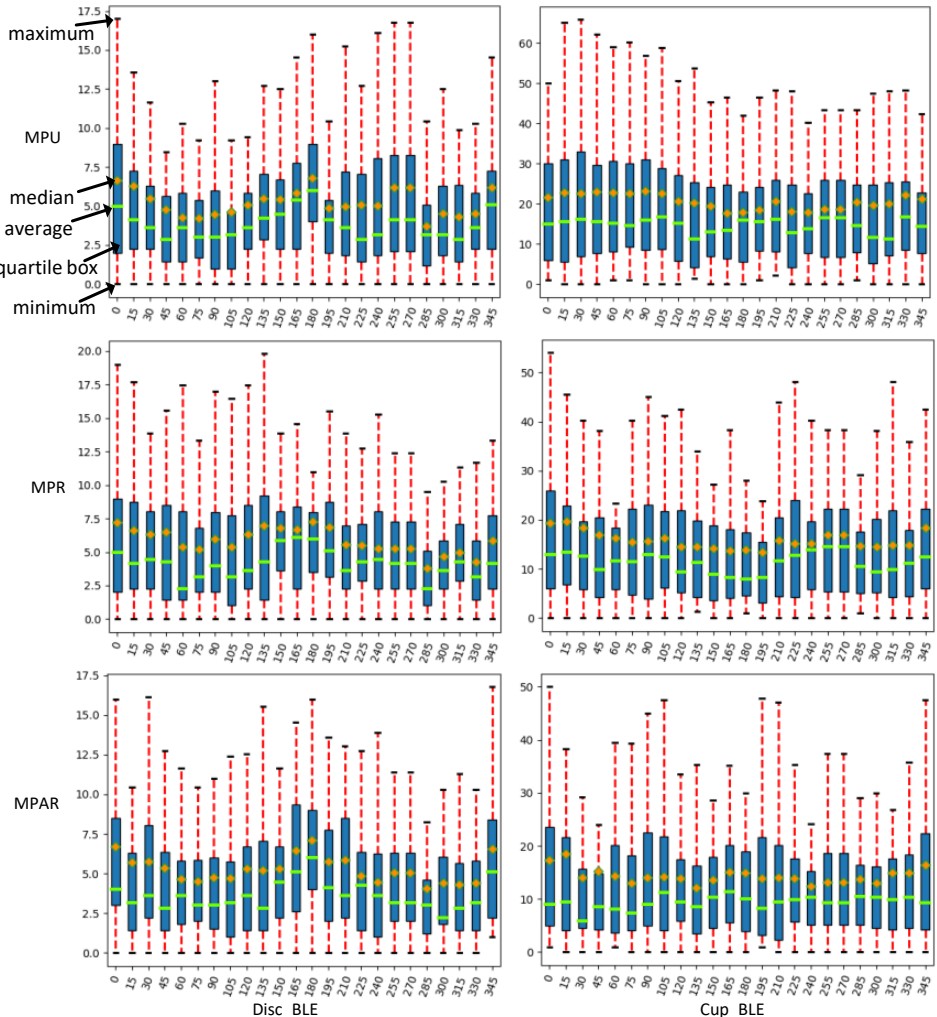

**Figure 6.** Box plot of BLE statistics for each model of the Drishti-GS1 dataset.

**Table 5.** Boundary distance localization error (BLE) comparison of model changes based on the Drishti-GS1 dataset.

| Model | Disc | Cup |
|-------|------|-----|
| MPU | 5.832/3.301 | 20.584/16.697 |
| MPR | 5.206/2.876 | 15.087/12.238 |
| MPAR | **4.765/2.933** | **14.342/10.169** |

This paper compares the receiver operating characteristic curve (ROC) curves of different models in the optic cup and joint optic cup segmentation, as shown in Figure 7. The MPR model has an increase in the area under the curve (AUC) of the optic disc by 0.0441 compared with the MPU model. The MUC model increased the AUC for the optic cup by 0.01 compared to the MPR model. The AUC of the MPAR model for joint optic disc segmentation was 0.9957.

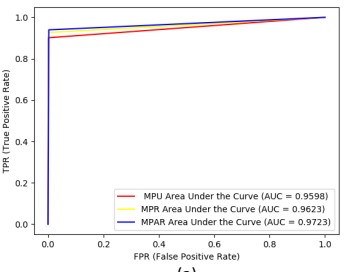 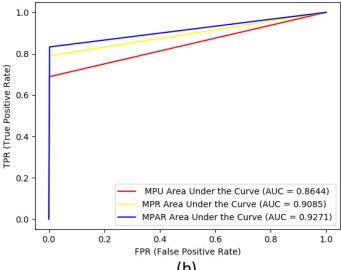 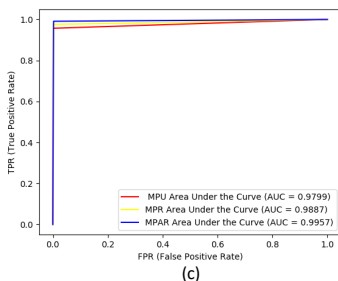

**Figure 7.** Comparison of receiver operating characteristic (ROC) curves of each model in the Drishti-GS1 dataset. (**a**) Cup; (**b**) Disc; (**c**) Optic.

In order to further illustrate the effectiveness of this method, this paper conducts a generalization verification on the DRIVE dataset. Table 6 shows the segmentation results of different models, which can show the effectiveness of the multipath U-Net in retinal vessel segmentation. It can also be seen that it is reasonable to combine convolutional neural networks and recurrent neural networks. At the same time, the results also show that the addition of an attention gate to the recurrent block in this paper also improves the performance of blood vessel segmentation.

**Table 6.** Comparison of model changes based on the DRIVE dataset.

| Model | Accuracy | Specificity | Sensitivity | F1 |
|-------|----------|-------------|-------------|-----|
| MPU | 0.9564 | **0.9835** | 0.8022 | 0.8196 |
| MPR | 0.9599 | 0.9823 | 0.8145 | 0.8207 |
| MPAR | **0.9642** | 0.9820 | **0.8173** | **0.8275** |

Figure 8 is an effect diagram of the blood vessel segmentation by each model. As shown in the figure, the MPAR model can effectively segment the small peripheral blood vessels effectively. and at the same time, it can more completely segment the blood vessels, reducing the problem of blood vessel rupture during segmentation. Figure 9 shows the ROC curve of each model on the DRIVE data set. Comparing the MPA model and the MPU model, the AUC area increased by 0.0034, and the MPAR model AUC area was 0.9754.

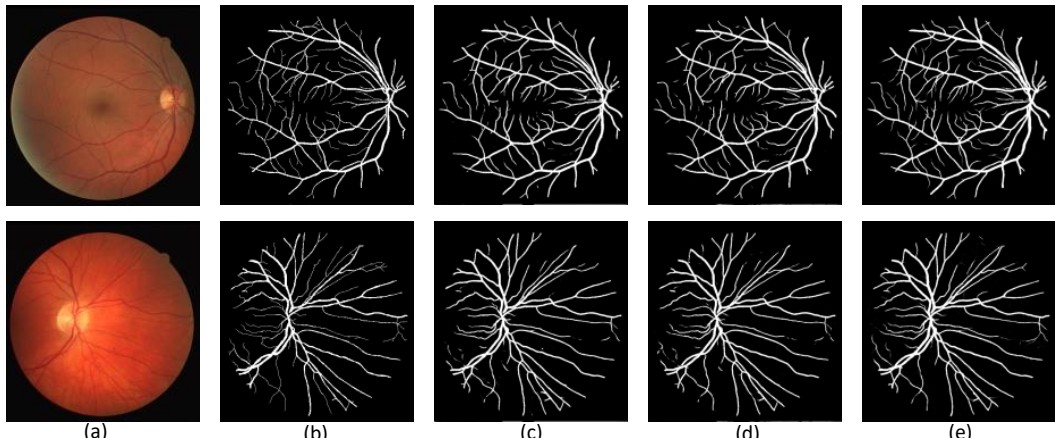

**Figure 8.** Comparison of segmented images of each model in the DRIVE dataset. (**a**) is original image; (**b**) is ground truth of (**a**); (**c**) MPU; (**d**) MPR; (**e**) MPAR.

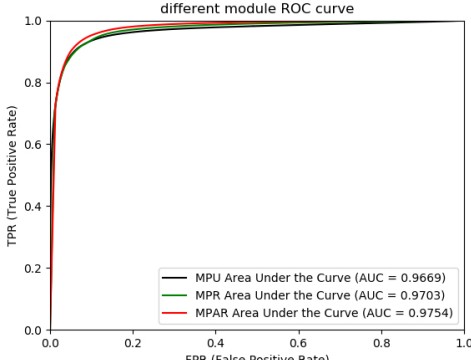

**Figure 9.** Comparison of ROC curves of the DRIVE dataset model.

## 4.5. Comparison of Results of Different Segmentation Algorithms

This paper compares the performance of the proposed model with other segmentation methods: the modified Chan-Vese model [43], super-pixel method [44], multi-field depth discontinuity method [45], U-Net [22], BCRF [46], multi-feature-based method [47], improved U-Net [22], CGAN [48] and deep learning-based methods [49]. Five evaluation indexes were compared. A value of "-" indicates that the results of this evaluation index were not given in the original work, as shown in Tables 7 and 8.

Although a large number of studies did not calculate accuracy, specificity and sensitivity, as shown in Table 7, compared with the results given in [22,24,48], the proposed method achieves optimal accuracy and specificity of the OD segmentation. Table 8 also shows the comparison of [24,49]. The method in this paper achieves the best accuracy and specificity of segmentation of the OC. The indicators calculated by a large number of works are Dice and BLE; Tables 7 and 8 show that, in these two evaluation indicators, the proposed method is superior to other OD and OC segmentation algorithms. This shows the benefit of expressing the segmentation task as a bounding box detection problem in our work.

**Table 7.** Comparison of the OD results of different baseline methods on the Drishti-GS1 dataset.

| Model | Accuracy | Specificity | Sensitivity | Dice | BLE |
|---|---|---|---|---|---|
| Vessel Bend [43] | - | - | - | 0.9600/0.0200 | 8.930/2.960 |
| Superpixel [44] | - | - | - | 0.9500/0.0200 | 9.380/5.750 |
| Multiview [45] | - | - | - | 0.9600/0.0200 | 8.930/2.960 |
| U-Net [24] | 0.9700/- | 0.9800/- | 0.9600/- | 0.9500/- | - |
| BCRF [46] | - | - | - | 0.9700/0.0200 | 6.610/3.550 |
| Gao et al. [47] | - | - | - | 0.9470/- | 8.885/- |
| Artem et al. [22] | 0.9353/- | 0.9684/- | 0.8623/- | 0.9500/- | - |
| CGAN [48] | 0.9800/- | 0.9900/- | **0.9800**/- | 0.9700/- | - |
| Xiao et al. [49] | - | - | - | 0.9400/- | 9.883/- |
| MPAR (ours) | **0.9967/0.0022** | **0.9981/0.0022** | 0.9365/0.0659 | **0.9817/0.0156** | **4.765/2.933** |

**Table 8.** Comparison of the OC results of different baseline methods on the Drishti-GS1 dataset.

| Model | Accuracy | Specificity | Sensitivity | Dice | BLE |
|---|---|---|---|---|---|
| Vessel Bend [43] | - | - | - | 0.7700/0.2000 | 30.510/24.800 |
| Superpixel [44] | - | - | - | 0.0800/0.1400 | 22.040/12.570 |
| Multiview [45] | - | - | - | 0.7900/0.1800 | 25.280/18.000 |
| U-Net [24] | 0.9700/- | 0.9800/- | **0.9600**/- | 0.8500/0.100 | 19.530/13.980 |
| BCRF [46] | - | - | - | 0.8300/0.1500 | 18.610/13.020 |
| Gao et al. [47] | - | - | - | 0.8260/- | 21.980/- |
| Xiao et al. [49] | 0.9712/- | 0.9758/- | 0.8598/- | - | - |
| Niharika et al. [50] | - | - | - | 0.8110/- | 23.335/- |
| MPAR (ours) | **0.9950/0.0022** | **0.9980/0.0020** | 0.9365/0.0639 | **0.8921/0.0793** | **14.342/10.169** |

Figure 10 shows the effect of the video cup segmentation of the model and other methods in this paper. Compared with columns d, f, g, and h, the proposed method is more accurate at segmenting the optic disc and the boundary is more reasonable, especially compared with the Graph Cut prior method [51]. Compared with column e, the segmentation boundary of the superpixel method is a relatively regular ellipse. This method is affected by the conventional shape. The extracted boundary results are not accurate and reliable. This also proves that the model proposed in this paper can more accurately and effectively collect the boundary information of the OD and OC.

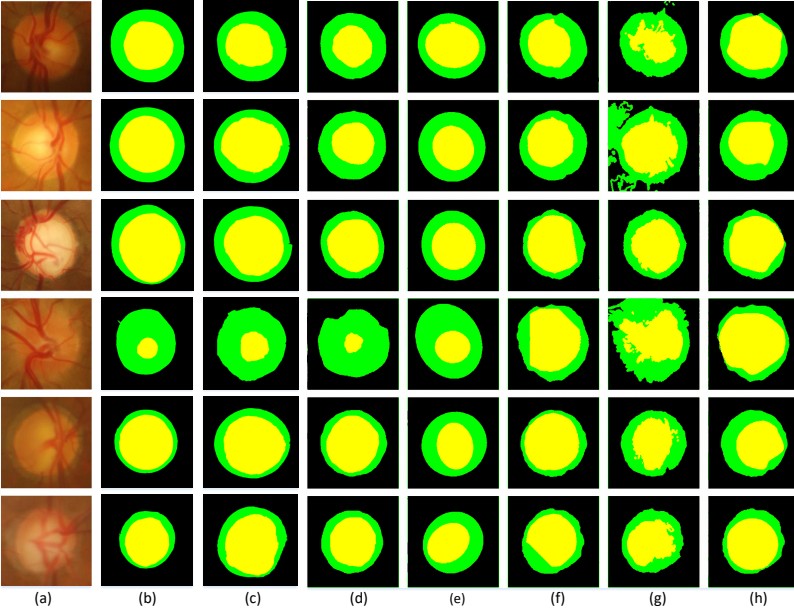

**Figure 10.** Comparison of different method model segmentation images of the Drishti-GS1 dataset. (**a**) is the original image; (**b**) is the ground truth; (**c**) is the result of the method in this paper; (**d**) BCRF [46]; (**e**) Superpixel [44]; (**f**) Multiview [45]; (**g**) Graph Cut prior [51]; (**h**) Vessel bend [43].

The segmentation of retinal blood vessels is challenging because retinal blood vessels are too small and diverse in shape. In order to verify the effectiveness of the method in this paper, we carry out a push verification on the DRIVE dataset and compare it with several recent methods, as shown in Table 9. The accuracy of the vascular segmentation and F1 score of the method in this paper reached optimal values, which were 0.9642 and 0.8275, respectively. The specificity is equal to that of the method in [52], and the sensitivity is only 0.0109 smaller than that of the method in [53], but 0.0747 more than that of the method in [54]. The segmentation effect is shown in Figure 11. This article enlarges part of the peripheral blood vessels. It can be seen that the method in this paper is more accurate than the method in [11,53] in segmenting small blood vessels and can effectively reduce the problem of small blood vessel rupture. In summary, the MPAR model in this paper can detect objects with variable sizes better than other methods.

**Table 9.** Comparison of model changes based on the DRIVE dataset.

| Model | Accuracy | Specificity | Sensitivity | F1 |
|---|---|---|---|---|
| Yan et al. [52] | 0.9538 | 0.9820 | 0.7361 | - |
| U-Net [24] | 0.9554 | 0.9802 | 0.7849 | 0.8175 |
| R2U-Net [11] | 0.9556 | 0.9813 | 0.7792 | 0.8171 |
| Samuel et al. [53] | 0.9609 | 0.9738 | **0.8282** | - |
| Chen et al. [54] | 0.9453 | 0.9735 | 0.7426 | - |
| Wang et al. [55] | 0.9567 | 0.9816 | 0.7940 | 0.8270 |
| MPAR (ours) | **0.9642** | **0.9820** | 0.8173 | **0.8275** |

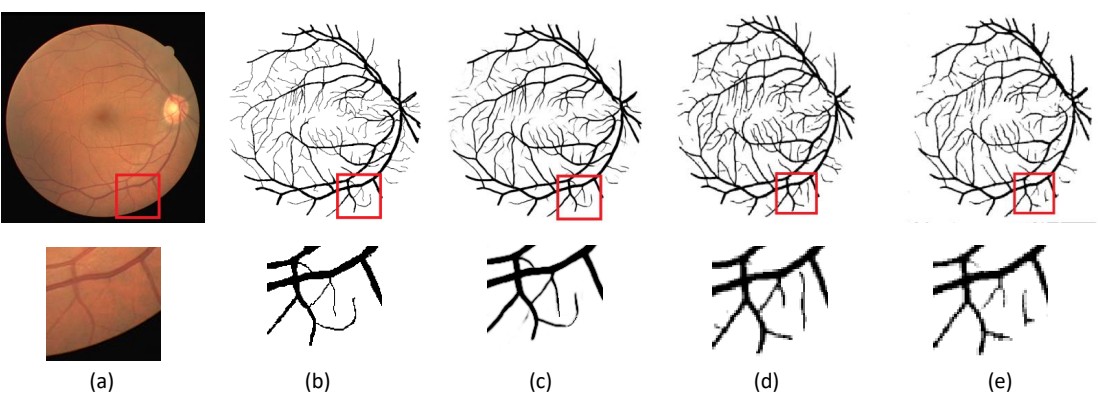

| (a) | (b) | (c) | (d) | (e) |

**Figure 11.** Comparison of model segmentation images of different DRIVE dataset. (**a**) is the original image; (**b**) is the ground truth; (**c**) is the result of the method in this paper; (**d**) method in [11]; (**e**) method in [53].

## 5. Conclusions

The analysis of medical fundus images has significance in terms of the effects of ophthalmic diseases. In this article, we propose a deep learning architecture called multi-path recurrent U-Net. This structure combines a recurrent neural network and a convolutional neural network and makes corresponding improvements to the original U-Net and the recurrent unit. It implements a multi-path recurrent U-Net architecture and improves the problem of disc division and cup division. Through experimental verification, more efficient segmentation results are shown to be generated for the Drishti-GS1 dataset. At the same time, the method was popularized and verified on the retinal blood vessel segmentation dataset DRIVE and has achieved good results. In this paper, our method was validated on 2D images; its extension to 3D data would be a possible future development.

**Author Contributions:** Methodology, F.W.; Software, F.W.; Supervision, Y.J.; Validation, J.G.; Writing-original draft, F.W.;Writing-review and editing, F.W. and S.C. All authors have read and agreed to the published version of the manuscript.

**Funding:** This work was supported in part by the National Natural Science Foundation of China (61962054), in part by the the National Natural Science Foundation of China (61163036), in part by the 2016 Gansu Provincial Science and Technology Plan Funded by the Natural Science Foundation of China (1606RJZA047), in part by the 2012 Gansu Provincial University Fundamental Research Fund for Special Research Funds; Gansu Province Postgraduate Supervisor Program in Colleges and Universities (1201-16), in part by the Northwest Normal University's Third Phase of Knowledge and Innovation Engineering Research Backbone Project (nwnu-kjcxgc-03-67).

**Conflicts of Interest:** The authors declare no conflict of interest. The funders had no role in the design of the study; in the collection, analyses, or interpretation of data; in the writing of the manuscript, or in the decision to publish the results.

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
