# Peer review of "Multi-Path Recurrent U-Net Segmentation of Retinal Fundus Image"

_applsci, doi:10.3390/app10113777_

Round 1

Reviewer 1 Report

The paper titled “Multi-path Recurrent U-Net Segmentation of Retinal Fundus Image” presents a multi-path recurrent U-Net architecture for OD, OC and retinal vessel segmentation. The comparison with existing methods show that the proposed architecture achieves state-of-the-art results. This manuscript can be published to the Applied Sciences journal after making minor corrections.

1. l. 1-2: This sentence must be rephrased.

2. l. 4: replace “prevent” by “identify”.

3. l. 4: “Aiming at the problems that the” must be removed from the sentence.

4. l. 9-10: At the same time, our method achieves state-of-the-art results in the segmentation of the Drishti-GS1 dataset.

5. l. 22- 24: This sentence is too long and should be rephrased.

6. l. 32. Deep Convolutional Neural Networks (DCNN) show …

7. l. 33-34. This sentence is too abstract. Authors should give a brief description of architecture introduced by [9].

8. l. 73. What does MPRU mean? When using an acronym for the first time, the full name should be provided.

9. l. 75. Authors should rephrase this sentence. “...the method for the segmentation of the OD and OC achieved the state-of-the-art results.

10. l. 76. remove “popularized and” for the sentence.

11. l. 88. The authors should define the acronym FCN.

12. l. 90. Authors should rephrase the “Since the U-Net effect is really good, …” part. For example, “Since the U- Net achieves acceptable results.”

13. l. 111-112. Recurrent neural networks have … Therefore, they are widely used ….”

14. l.131. “The proposed architecture is based…” instead of “This paper is based..”. Similarly, authors must replace “The network structure in this paper...” with “The proposed network structure…”.

15. l. 134. Please, remove the “of this paper”.

16. l. 153. and l. 154. “This paper...” must be replaced by “The proposed method...”. Similar corrections must also be made in the remaining manuscript (lines 164, 167, 169, 170, 194- 195).

17. RGU instead of AGU.

18. l. 211. The methods was built on the open source deep learning library PyTorch….

19. Tables 1, 2, 3 4, 5, 6, 7 and Figures 5, 6, 7, 8 and 9. The acronyms of the methods must be corrected. Similar corrections must also be made in the remaining manuscript. The proposed method is named as MPRU in the abstract and conclusions sections, however this acronym is not met in the manuscript.

20. Figure 6. The spacing of the x -axis labels must be increased.

21. l. 287, 288. The authors should remove ‘the Literature’ and add the corresponding methods’ names. The remaining manuscript requires similar corrections.

22. l. 294. ...has significance and effects...

Author Response

Dear reviewer, Thank you very much for your careful and professional comments on my manuscript. The attachment is my answer to your question. If you have any questions, please contact me in time. I wish you good health and a happy life. Kind regards, Falin Wang

Reviewer 2 Report

The manuscript by Jiang et. al. proposed a multi-path recurrent U-Net construction to segment optical disk and optical cup of human eyes, and also verified the proposed method on retinal blood vessel segmentation. The manuscript is presented in a clear manner. However, the manuscript needs significant improvements before it can be considered for publication in Applied Science. Here are my major and minor concerns and I would appreciate it if the authors can further clarify on or improve these issues.

Major comments:

  1. In Introduction, the authors spent long paragraphs simply listing or summarizing a lot of other people’s works without comparing or discussing their pros and cons. Also, the motivation to propose the MPAR and its relevance to all previous works are poorly discussed. This part needs significant improvements.
  2. For training and test: “The Drishti-GS1 dataset contains a total of 101 fundus images, 31 normal and 70 diseased. The training set contains 50 images and the test set contains 51 images”. How did the authors split training and test sets? Did the authors balance normal and diseased data while doing data augmentation in training? Can the authors quantitatively show more details about these?
  3. For tables:

There are two values for each result in Table 1-3, but only one for each result in Table 4. What are their differences? If the second values of each result in Table 1-3 are standard deviations, the authors need to calculate the p values to statistically justify that the newly proposed model really has better performance than the other two. Otherwise the authors cannot conclude the proposed MPAR model is better than MPU and MPR.

  1. For AUC curves:

How many test data are Figure 7 and Figure 9 calculated from? I suppose they are not generated from a single test image, but the mean AUC of all test data. If this is the case, can the authors provide the corresponding AUC distributions and standard deviations of all test data in Figure 7 and Figure 9, respectively, probably in appendix?

Minor comments:

  1. For figures:
  • The authors need to clarify the full name of BN in Figure 3(b); the sub-figure alphabets (should be a-e) are messed up in Figure 3.
  • What do the numbers (_011 and _22) and GT mean in Figure 4?
  • The authors need to add legends and axis labels in Figure 6
  • The font size and type of all figures in this manuscript should be uniform
  1. Typos: Line 169, Ht; Line 235, what is MPA? Line 244, what do 51 and 24 mean? There are other typos which are not listed here. There should be a grammar and typo check of this manuscript. Also, the writing of the manuscript should be condensed.

Author Response

(The authors gave the same response as above.)

Reviewer 3 Report

Dear Authors,

below you may find my comments.

The idea proposed in the paper is promising. Also, proposed method, as well as simulations are presented in detail. However, there are some issues in the paper that should be tackled.

1. Abstract
Authors should avoid using accronyms and abbreviations in the abstract. I suggest authors to remove OD, OC, MPRU from the abstract.
Also, abstract should be slightly extended to include more details regarding the performed experiments and simulations.

2. Introduction should more clearly define the research objective (aim) and the research question. Also, some sentences should be reformulated, for example: "The main works of this article are...".
Also, as the part of the literature review more papers from this domain should be added. For example, since it is a very promising field, it should be mentioned that the swarm algorithms have successfully been applied hybridized with CNNs. You may consider using the following papers in the literature review:

Nebojsa Bacanin, Timea Bezdan, Eva Tuba, Ivana Strumberger, Milan Tuba, Optimizing Convolutional Neural Network Hyperparameters by Enhanced Swarm Intelligence Metaheuristics, Algorithms, Vol. 13, Issue 3, pp.67

de Rosa, G.H.; Papa, J.P.; Yang, X.S. Handling dropout probability estimation in convolution neural
818 networks using meta-heuristics. Soft Computing 2018, 22, 6147—-6156.

3. Conclusion
Conclusion section should be extended to include more details regarding the authors' future work.

4. References
Some references have missing parts, like the pp., publisher, etc. Also, references list should be updated to include more literature sources that are relevant.

5. English language/other
There are some English spelling and grammar errors. Also, some sentences should be reformulated.

Author Response

(The authors gave the same response as above.)

Round 2

Reviewer 2 Report

I think the authors have addressed my comments and it's eligible for publication. 

However, I strongly recommend the following for both the requirements of any peer-reviewed journal and the authors' own publication quality (once it is published, any typos, grammar mistakes, ununiform figure font type, font size and mismatch image resolutions can no longer be changed, and we all want our work to look decent and beautiful):

  1. Please have a thorough check of the English grammar and writing (there still exist typos) 
  2. Please uniform the font size, font type, and image resolution of all figures in the manuscript as much as you can. At least make it visually decent. 

Author Response

Dear Reviewer,

Thank you very much for your revision of my manuscript. I have carefully checked and revised my manuscript grammar and writing errors. At the same time, I also improved and modified the images in the manuscript. If you have any other questions about the manuscript, please contact me in time and I will respond to you as soon as possible. I wish you good health and happy work.

Best Regards,

Falin Wang